



# Estimation of missing building height in OpenStreetMap data: a French case study using GeoClimate 0.0.1

Jérémy Bernard[1,3], Erwan Bocher[2], Elisabeth Le Saux Wiederhold[3], François Leconte[4], and Valéry Masson[5]

[1]University of Gothenburg, Department of Earth Sciences, Sweden
[2]CNRS, Lab-STICC, UMR 6285, Vannes, France
[3]Université Bretagne Sud, Lab-STICC, UMR 6285, Vannes, France
[4]Université de Lorraine, INRAE, LERMaB, F88000, Epinal, France
[5]Météo-France and CNRS, CNRM, UMR3589, Toulouse 31057, France

**Correspondence:** Jérémy Bernard (jeremy.bernard@zaclys.net)

**Abstract.** Information describing the elements of urban landscape is a required input data to study numerous physical processes (e.g climate, noise, air pollution). However, the accessibility and quality of urban data is heterogeneous across the world. As an example, a major open-source geographical data project (OpenStreetMap) demonstrates incomplete data regarding key urban properties such as building height. The present study implements and evaluates a statistical approach which models the missing values of building height in OpenStreetMap. A Random Forest method is applied to estimate building height based on building's closest environment. 62 geographical indicators are calculated with the GeoClimate tool and used as independent variables. A training data set of 14 French communes is selected, and the reference building height is provided by the BDTopo IGN. An optimized Random Forest algorithm is proposed, and outputs are compared with an evaluation dataset. At building scale for all cities, at least 50% of the buildings have their height estimated with an error being less than 4 m (the city median building height ranges from 4.5 m to 18 m). Two communes (Paris and Meudon) demonstrate building height results out of the main trend due to their specific urban fabric. Putting aside these two communes and when building height is averaged at regular grid scale (100m × 100m), the median absolute error is 1.6 m and at least 75% of the cells of any city have an error lower than 3.2 m. This level of magnitude is quite reasonable when compared to the accuracy of the reference data (at least 50% of the buildings have an height uncertainty equal to 5 m). This work offers insights about the estimation of missing urban data using statistical method and contributes to the use of open-source data set based on open-source software. The software used to produce the data is freely available at https://zenodo.org/record/6372337 and the data set can be freely accessed at https://zenodo.org/record/6396361.

## 1 Introduction

The topography - defined as the spatial distribution of natural and artificial land use features - has a significant influence over the microclimate. This is clearly visible in urban areas where the great heterogeneity of forms, materials and land uses, induces a high variability of temperature, wind speed and humidity (Oke, 2002). Thus an in-depth knowledge about the topography of



a location would lead to understand better and to model more accurately its climate but also other physical processes such as noise propagation and air pollution (Tang and Wang, 2007; Bocher et al., 2019).

There is currently no standard geographical data to study the urban climate worldwide. However, urban data tend to increase both under closed licence and open-source. A key open-source data approach is the OpenStreetMap (OSM) [1] project. Data from the latter have several features: they are expected to be available worldwide and the most important objects needed for urban climate studies (building footprints, isolated tree locations, water, vegetation and impervious patches) are available, well located and described using a great diversity of tags (Mocnik et al., 2017). Moreover, OSM has a free tagging system that allows users to improve the current tags (key, value) with their own information (an OSM user can describe a building object with the tags such as the following: "building"= "house", "height"="10", "building:levels":"2").

However, information concerning the vertical dimension is rarely available (Masson et al., 2020). Lao et al. (2018) reported that less than 3% of the buildings globally have an height value and less than 4% have a number of levels value. For the city of Paris (where this information is known as quite well informed), the values are only 0.1% and 51.2%, respectively. This is a major shortcoming since the urban climate is often characterized by spatial indicators based on the third dimension:

- the Sky View Factor (SVF), which is calculated according to terrain level variations, building and tree locations and heights, is related to effective albedo (Bernabé et al., 2015), strongly correlated to temperature (Lindberg, 2007) and wind speed (Johansson et al., 2016),

- the building height variability within an area affects the vertical and horizontal wind speed (Hanna and Britter, 2010),

- the roughness length of an area, often calculated using facade density, is used to estimate the urban canopy wind speed vertical profile (Hanna and Britter, 2010).

The objective of this study is to develop a method to estimate the height of a building from its topographical context using only data available in OSM. Modelling building footprints and their height value have been largely covered by remote sensing. It can cover large areas at once quite efficiently and the resulting data sets can be updated quite easily with a repetitive coverage. Different techniques for building height extraction have been developed, based on photogrammetric processing (Fradkin et al., 1999; Zeng et al., 2014), analysis of point clouds from airborne light detection and ranging (Sohn et al., 2008; Shan and Toth, 2018), shadow detection (Song et al., 2013; Shao et al., 2011) and more recently deep learning approach (Cao and Huang, 2021). In the mean time, the recent and global movement on open data, specifically for vector topographic databases such as OSM, offers new opportunities to estimate the building height. The geography of a territory, the pattern of the topographic elements are criteria that can be used as a proxy to identify the urban forms and therefore the distribution of the building heights. However, the literature concerning such method is scarce: at our knowledge, only Biljecki et al. (2017) have used building footprints and their corresponding attributes to derived the building height. They have tested several random forest models using as independent variables building properties characterizing its geometry footprint (size, shape and number of neighbors), other attributes (use, age and number of levels), and informations concerning the inhabitants (the number and their

---

[1]www.openstreetmap.org





level of income). The method presented in the next sections use the same type of approach but using only OSM data. However,

OSM data do not contain as much detailed information about buildings as Biljecki et al. (2017) had (number of levels, age, number of inhabitants) but other informations describing the environment of the buildings will be used (roads, vegetation, rail, etc.).

In order to make OSM data available to urban climate researchers, the GeoClimate tool has been developed (Bocher et al., 2021b, a). It is an easy way (**i**) to download most of the information needed for urban climate studies, (**ii**) to estimate building

height from the topographical context, (**iii**) to calculate spatial indicators (such as SVF, building height variability or roughness length) useful as input for parametric climate models. This paper focuses on the second item, namely how to estimate building height in OSM when the information is missing. First, the data and the methodology used to estimate the building height is presented (Sect. 2) and second the accuracy of these estimations is analysed (Sect. 3).

## 2  Data and Method

This study presents a method to estimate values of building height when the information is missing in OSM. The height of a building is determined according to a regression based statistical model (e.g. Random Forest model) using a set of spatial indicators as independent variables, including the building's shape, the building's relation to its neighbours and the organization and morphology of the building's environment. The true building height values come from the BDTopo V2.2 (BDT) provided by the French National Geographic Institute (IGN). These values are defined as reference height. Two data sets have been

considered for this study, namely a training data set to build the Random Forest algorithm and a validation data set to compare the outputs of the optimized Random Forest algorithm with reference heights. The overall methodology is illustrated in Figure 1 and consists in the following steps:

1. Building characterization: each building and its environment are characterized by spatial indicators (building area, number of buildings neighbors, vegetation fraction, etc.). These indicators are the independent variables of the statistical

model.

2. Building height attribution: the reference building heights (BDT building heights) are attributed to each OSM building according to their footprints. The resulting height in the OSM dataset is the dependent variable of the statistical model.

3. Statistical analysis: the Random Forest model is built based on the training data set. In this step, parameters that maximise the performance of the Random Forest model are identified.

4. Performance evaluation: outputs of the optimized Random Forest model are evaluated against the reference heights of the validation data set.

Each step is described further in Sect. 2.2.





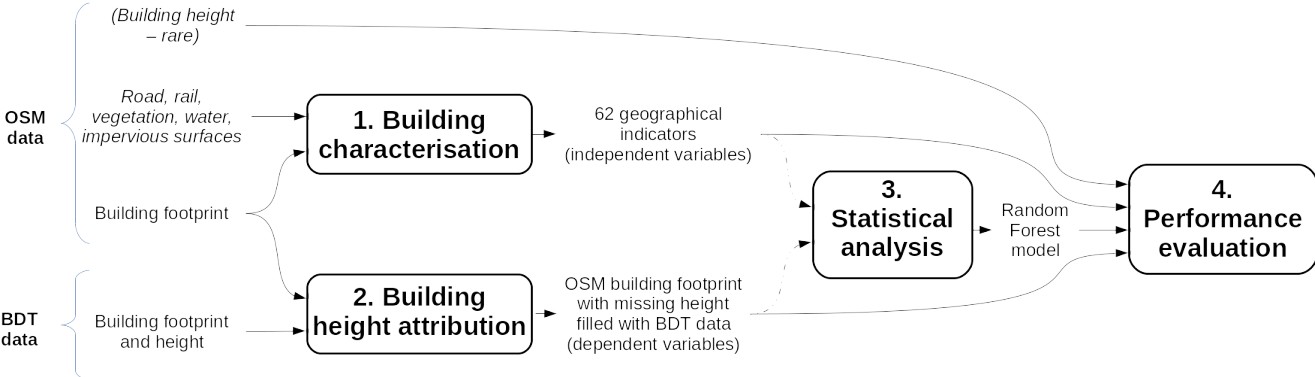

**Figure 1.** Overall methodology - the use of dashed arrows means that only the training data set is used

## 2.1 Study area

Building organization and height may differ a lot throughout the world, limiting the ability to model the height of a building
based on the characteristics of its environment. Thus, although the method can be used to estimate the height of any building
in the world, the application area of this preliminary work is limited to the French territory. The training and evaluation areas
are selected to cover all types of commune (from small villages to large conurbations). Thus the dataset contains communes
belonging to each of the four French commune types defined by the French National Institute of Statistics and Economic
Studies (INSEE, 2020): main urban area, secondary urban area, peripheral urban area, rural area. According to the French
2020 census data, the types are defined based on the following communes characteristics: number of inhabitants, density of
population, number of employees, population flow between households and workplaces. They are used to define the urban
attraction cluster. The definitions of each type is given Table 1.

| Commune type | Definition |
|---|---|
| Main urban area | Commune centre of the urban attraction cluster |
| Secondary urban area | Other commune of the urban attraction cluster |
| Peripheral urban area | Commune in the attraction area of the urban cluster |
| Rural area | Commune outside from any urban attraction area |

**Table 1.** The four commune types defined by INSEE (2020)

The training and the evaluation data sets contain respectively 14 and 8 communes. The location of each commune is shown
Figure 2 while further information concerning each territory is given Table 2 and Table 3 (respectively for training and evalua-
95 tion data sets).





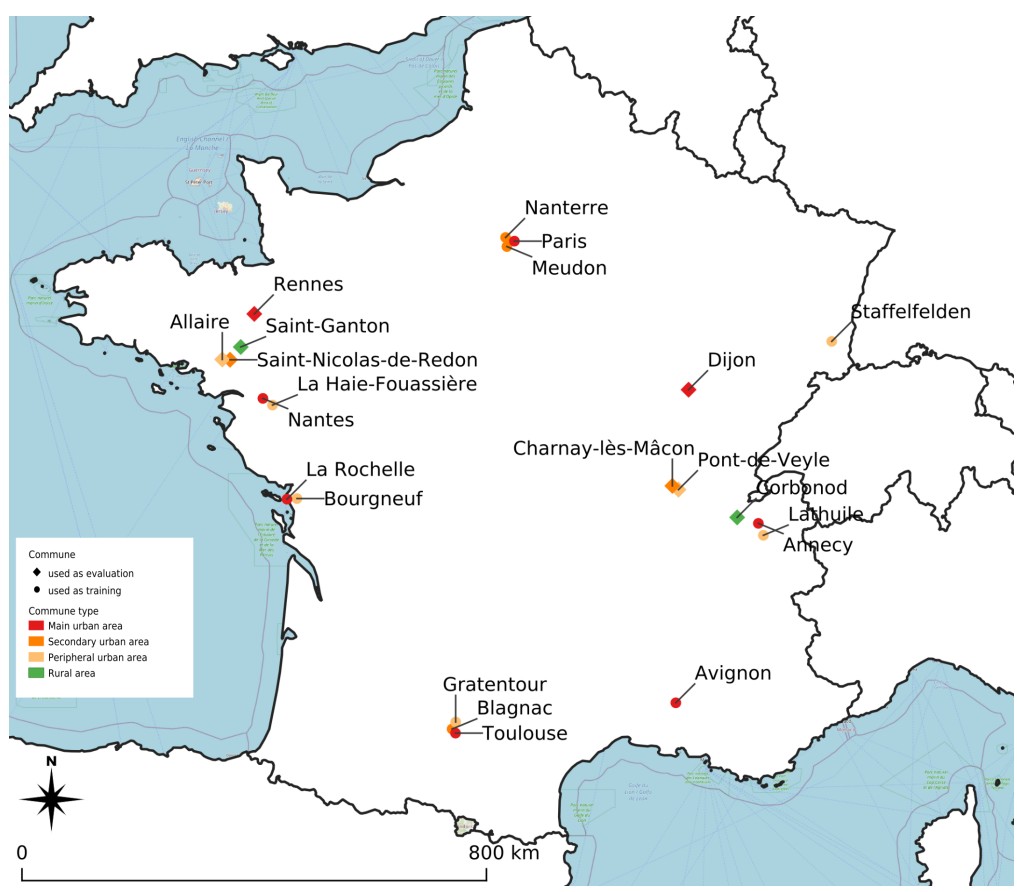

**Figure 2.** Location of the 22 communes used as training or evaluation data. ©OpenStreetMap contributors 2021. Distributed under the Open Data Commons Open Database License (ODbL) v1.0.





| Commune type | Commune name | Inhabitants (2017) | Number of buildings* | INSEE code |
|---|---|---|---|---|
| Main urban area | Paris | 2187526 | 15964 | 75056 |
| | (6th, 11th and 18th districts) | (40525, 145903, 193665) | | (75106, 75111, |
| | Toulouse | 479553 | 103368 | 31555 |
| | Nantes | 309346 | 57550 | 44109 |
| | Annecy | 126924 | 21153 | 74010 |
| | Avignon | 91921 | 29113 | 84007 |
| | La Rochelle | 75735 | 31194 | 17300 |
| Main peripheral urban area | Nanterre | 95105 | 10851 | 92050 |
| | Meudon | 45352 | 5430 | 92048 |
| | Blagnac | 24517 | 9286 | 31069 |
| Secondary peripheral urban area | La Haie Fouassiere | 4659 | 2323 | 44070 |
| | Gratentour | 4158 | 2938 | 31230 |
| | Staffelfelden | 3959 | 2254 | 68321 |
| | Bourgneuf | 1275 | 782 | 17059 |
| | Lathuile | 1016 | 732 | 74147 |

(*) only building having an height value higher than 3 m in the BDT (cf Sect. 2.2.2) have been conserved for this evaluation

**Table 2.** Information and statistics about the training data set

| Urban class | Commune name | Inhabitant (2017) | Number of buildings* | INSEE code |
|---|---|---|---|---|
| Main urban area | Rennes | 216815 | 30527 | 35238 |
| | Dijon | 156920 | 23044 | 21231 |
| Main peripheral urban area | Charnay-lès-Macon | 7376 | 3574 | 71105 |
| | Saint-Nicolas de Redon | 3179 | 2515 | 44185 |
| Secondary peripheral urban area | Allaire | 3854 | 2744 | 56001 |
| | Pont-de-Veyle | 1625 | 729 | 1306 |
| Rural area | Corbonod | 1264 | 1115 | 1118 |
| | Saint-Ganton | 424 | 469 | 35268 |

(*) only building having an height value higher than 3 m in the BDT (cf Sect. 2.2.2) have been conserved for this evaluation

**Table 3.** Information and statistics about the validation data set





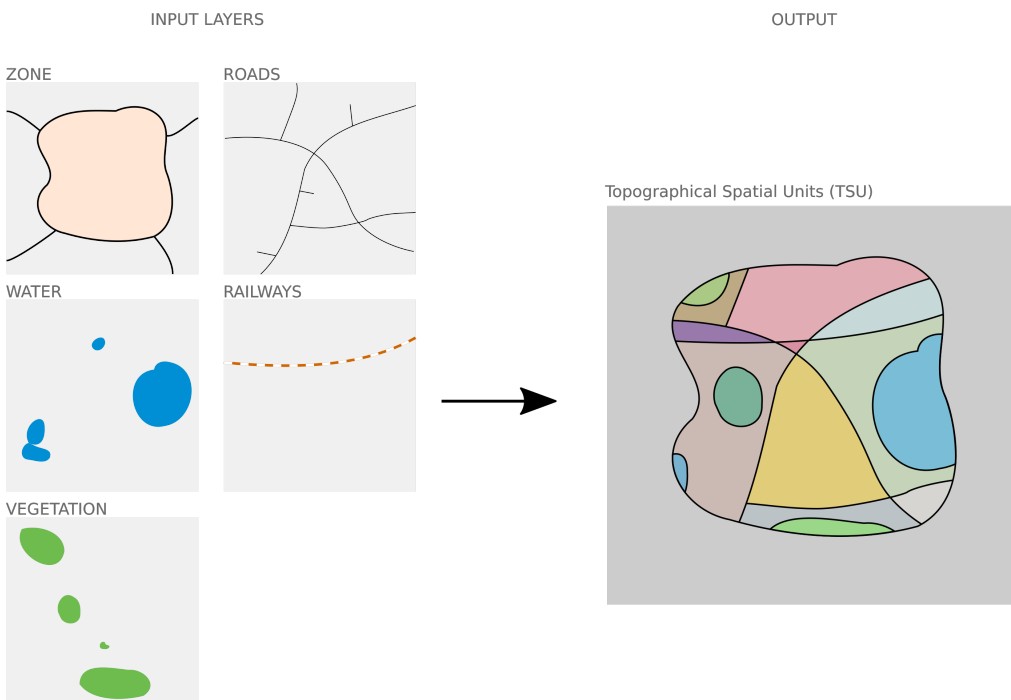

**Figure 3.** Example of Topographical Spatial Unit calculation

## 2.2 Methodology

### 2.2.1 Building characterization

Data from OSM is used to characterize the building and its environment: building footprint, vegetation footprint and type, water footprint, impervious footprint, rail and road footprint. The free and open-source GeoClimate software (Bocher et al., 2021b, a) is used to compute the spatial indicators at three different scales:

- building scale,

- block scale: it is defined as the aggregation of all buildings touching each other,

- Topographical Spatial Unit (TSU) scale: it is defined according to roads and rails central lines, commune boundaries, water and vegetation boundaries when their area is respectively higher than 2500 and 10000 m$^2$ (Figure 3).

Each building is described by a total of 62 spatial indicators (note that all characteristics are calculated only in two dimensions since most OSM data do not have height information). Four types of indicators are used (cf. Table 4) and the list of all indicators is given in Annex A. Indicators values calculated at block and TSU scales are then attributed to each building (a building within a given block or TSU embeds the indicator values of the block and the TSU it belongs to).





| Indicators type | Scale of application | | | Examples of indicators |
|---|---|---|---|---|
| | building | block | TSU | |
| Type and use | x | | | building type, building use |
| Form and size | x | x | x | area, form factor, fraction of courtyard, etc. |
| Spatial relations | x | | | minimum distance to an other building, fraction of wall shared with other buildings, minimum distance to road, etc. |
| Planar density | | | x | building fraction, vegetation fraction, etc. |
| Aggregated statistics from lower scale | | x | x | mean building area, standard deviation building form factor, etc. |

**Table 4.** Types of spatial indicators used to define the main characteristics of each unit scale

### 2.2.2 Reference building heights attribution

In order to train and evaluate the statistical model, a reference height used as true value shoud be assigned to each OSM buildings. Most of the OSM buildings do not have any information concerning their height. The few buildings with height information could have been considered as training and evaluation data set, however these buildings are sparse and not representative of common buildings (since most of them are filled by OSM users because they are well-known buildings with specificities). Therefore, the training and evaluation data set are created only with OSM buildings having no height. The ref-

erence height is set using the BDT data. However, a single building in OSM may match with several buildings in the BDT (Figure 4).

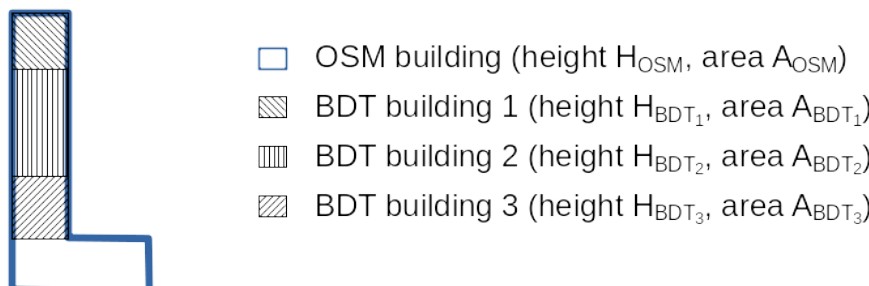

**Figure 4.** Example of overlap between OSM and BDT buildings

Thus, the height of an OSM building used as reference ($H_{osm,true}$) is calculated from the height of all intersected BDT buildings according to Eq. (1). This equation is applied for both the training and the validation data sets.

$$H_{osm,true} = \frac{\sum_{i=1}^{n} A_i \cdot H_{BDT_i}}{\sum_{i=1}^{n} A_i} \tag{1}$$





with $A_i$ the area of the intersection between an OSM building and a BDT building $i$, and $H_{BDT_i}$ the height of the BDT building $i$ intersecting the OSM building.

In the example presented in Figure 4, if BDT building 1 is much taller than the others (BDT building 2 and 3), this information is lost (smoothed by the averaging) and could then lead to a bias in the learning process. To keep a track of this potential bias, a simple index is proposed to characterize the proportion of intersection between a BDT building and an OSM building.

This index, called uniqueness value ($UV$), is defined in Eq. (2):

$$UV = \frac{max_{i \in 1..n} A_i}{\sum_{i=1}^{n} A_i} \tag{2}$$

The uniqueness value considers only the BDT building that demonstrates the largest intersection area with a given OSM building. The higher the $UV$, the more unique the BDT building intersecting the OSM building. $UV$ is not impacted by the fraction of OSM building shared with other BDT buildings. If only one BDT building overlaps only a small fraction of an OSM

building, the uniqueness value will be 1.

### 2.2.3 Design and optimization of the Random Forest statistical model

For the statistical analysis, the OSM building height (reference height $H_{osm,true}$) is defined as the dependent variable while spatial indicators are defined as independent variables. Only the training data set (cf. Table 2) is used for this step. To obtain an optimal model, the methodology illustrated Figure 5 is applied.

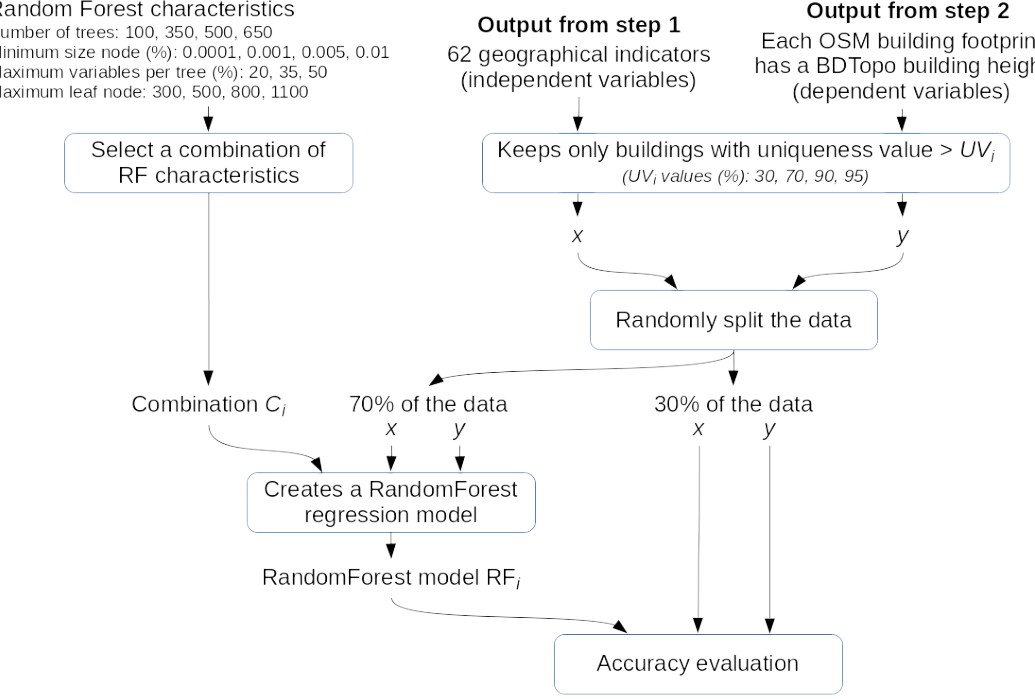

**Figure 5.** Method to train and optimize the Random Forest model. Only the training data set is used at this step.





The Random Forest (RF) approach is chosen for several reasons: (**i**) it is simple to implement, (**ii**) it deals with quantitative and qualitative variables and (**iii**) it is appropriate when using large number of variables (Hastie et al., 2001). In order to limit overfitting and high correlation between trees, all combinations of the following RF regressor parameters are investigated:

- number of trees: 100, 350, 500, 650 (note that preliminary analysis showed lower accuracy when the number of trees was lower than 100 and no significant improvement when greater than 650),

- minimum size node (minimum fraction of sample used to create a new node): 0.0001, 0.001, 0.005, 0.01% (the whole sample size includes 345418 individuals - note that preliminary analysis showed decreasing performance over 0.01 %),

- maximum variables per tree (maximum fraction of variables used in a tree): 20, 35, 50% (of a total of 62 variables - note that preliminary analysis showed decreasing performance when the fraction was lower than 20% and no significant improvement over 40%),

- maximum leaf nodes (maximum number of leaves in a tree): 300, 500, 800, 1100 (note that preliminary analysis showed no significant improvement over 1100 while increasing the complexity and thus potentially the overfitting).

For a default combination of 500 trees, 0.001% of minimum size node and 30% of maximum variables per tree, the effect of $UV$ on the accuracy is studied, keeping only buildings having a $UV$ above 30, 70, 90 and 95%.

70% of the training data is randomly drawn to construct the RF. The accuracy is calculated using the remaining 30% of 150   the data. This process is performed ten times for each combination $C_i$ and uniqueness value $UV_i$. The scikit-learn Python algorithm is used for this investigation.

The optimized combination $C_{opt}$ and uniqueness value $UV_{opt}$ leading to the lowest Mean Absolute Error (MAE) are used to construct the final RF model used in Geoclimate. For this purpose, the entire training dataset is used as input of the Smile library algorithm (since Geoclimate is Java-based).

### 2.2.4 Performance evaluation

The optimized RF model obtained at the previous step is run over the 8 communes of the validation data set to calculate the missing height values of the OSM buildings. For each building, the heights estimated with the optimized RF model ($\hat{H}_{OSM,model}$) are then compared to the reference height ($H_{OSM,true}$). The building height values filled by the OSM users ($\hat{H}_{OSM,user}$) are also compared to the reference height (actually if the user fills only the number of storey a simple rule is used 160   to calculate the building height). The model error $Err_{model}$ and the users error $Err_{user}$ are defined respectively for heights estimated by the Random Forest model (Eq. (3)) and heights estimated by the users (Eq. (4)):

$$Err_{model} = \hat{H}_{OSM,model} - H_{OSM,true} \tag{3}$$

$$Err_{user} = \hat{H}_{OSM,user} - H_{OSM,true} \tag{4}$$



In parametric urban climate models, parameters such as building height are aggregated within each square cell of a regular grid. Therefore, four indicators are calculated for a grid of 100 m width square: the mean and standard deviation building height, the roughness length (as defined by Hanna and Britter (2010)) and the SVF (as defined in Bernard et al. (2018)).

## 3    Results and discussions

The data set produced by the methodology described in section 2.2 can be freely downloaded at https://zenodo.org/record/6396361

(Bernard et al., 2021). In this section, cells having no building having their height estimated are not considered for the statistic calculations.

### 3.1    Optimized configuration of Random Forest characteristics

Very little accuracy difference is observed between all combinations described in Sect. 2.2.3. For all studied configurations, the median RMSE ranges between 2.05 m and 2.2 m. The minimum size node and the number of trees (when greater than 100) have

the least significant impact on the accuracy. The highest accuracy is reached when the maximum variables per tree is 50% and the maximum leaf nodes is 1100. Thus, the RF scenario chosen for Geoclimate has 350 trees, 40 % of maximum variables per tree (based on previous results showing little difference between 40% and 50%) to minimize the correlation between trees, 0.01 % of minimum size node and 1100 maximum leaf node to minimize the potential over-fitting (Hastie et al., 2001). Since the maximum tree depth obtained in Python for this configuration is 33, this value is also applied to the Geoclimate RF algorithm.

The uniqueness value has an unexpected effect on the accuracy: the MAE decreases when $UV$ value increases up to 70% and increases for $UV$ values above 90%, while it could be expected to continue decreasing. However, the difference is slight (0.05 m - 2.5 %) and may be explained by the size of the sample which is larger (+23%) for the 70% scenario than for the 95% (having respectively 345418 and 281081 individuals). Therefore, the data used to train the Geoclimate model is created with $UV = 70\%$.

### 3.2    General building height accuracy

For all cities, more than 50% of the buildings have an estimated height being within a +/- 3.97 m (3.22 m if the 18th Paris district is excluded) interval around the true building height (Figure 6a). At cell scale (Figure 6b), the same statistic is +/- 4.61 m (2.74 m if the 18th Paris district is excluded). If cities demonstrating a specific behaviour are not considered (Paris and Meudon), the median absolute error at cell scale is always lower than 1.6 m and 75% of the buildings or cells of any city have

an error lower than 3.2 m. This error is equivalent to one building floor's height and could appear quite high. However, it seems quite reasonable when compared with the accuracy of the reference data (i.e. the height uncertainty of more than half of the BDT data set is +/- 5 m).







**Figure 6.** Median absolute error versus median true value for (**a**) building height (**b**) RSU average building height (**c**) RSU standard deviation building height (**d**) RSU mean ground sky view factor and (**e**) RSU effective terrain roughness length. The cross and the dot are the medians while the whiskers are 1st and 3rd quartiles.





Surprisingly, the worst results are obtained with communes belonging to the training data set. Overall, there is almost no accuracy decrease when the model is applied to the validation data set (Figure 6). No city type shows a specific pattern: even

the rural areas which are not included in the training data set do not show a higher error than the overall trend. However, a specific behavior is observed for the city of Meudon and for the 18th Paris district which have a higher error than the main trend (Figure 6b). A part of the explanation can be found in their uncommon urban fabric which makes them more difficult to estimate by the RF model:

– Meudon has a quite low median height (Figure 6b) while a high building height variability within a 100m square (Figure

6c),

– the 18th Paris district has the highest average building roof height at grid scale (Figure 6b) but also the highest sky view factor of the three Paris districts (Figure 6d) while the lowest would be expected.

### 3.3 Accuracy of standard spatial indicators

The variability of height within a cell is very roughly calculated with the estimated height: for more than 50% of the cells,

the relative error on the standard deviation of the building height is higher than 50% (Figure 6c). This behavior is quite understandable since the RF model smooths the values of the estimated height: it cannot reproduce entirely the complexity of the initial data set. For more than 50% of the cells, the roughness length relative error is about 20%. This error is slightly higher for low roughness length and for the specific cases of Meudon and the 18th Paris district. The sky view factor seems the indicator the least affected by the building height estimations: for more than 50% of the cells, the relative error (calculated

with a SVF value of 1 as reference value) is almost always lower than 20% and sometimes reaches 10% - note that 75% of the cells have an absolute error lower than 0.07). The effect of the building height error is limited because the sky view factor is a 3-dimensional indicator: it also accounts for the horizontal footprint of the building which is the same between the estimated and observed data.

### 3.4 Limitations of the model for high-rise buildings

The accuracy differs a lot between low-rise and high-rise buildings. For all types of city, the building height is often overestimated for buildings smaller than 5 m and often underestimated for the taller ones (Figure 7). The bias for high-rise buildings can be quite high but it does not affect the general accuracy of the model since most of the buildings are low-rise (cf Figure 7: 80% of the buildings are lower than 10 m - even in main cities). A better estimation of the high-rise buildings may be achieved using a training data set containing an equal number of buildings for all levels. This would allow a better representation of

the spatial heterogeneity of the third dimension. However, this would most probably affect the accuracy of the estimation for low-rise building. Indeed, in France, low-rise building are much more numerous than the high-rise ones.

As previously observed, there is almost no accuracy decrease between the training and the validation estimations, even for high buildings. Only a slight difference can be observed for main urban cities: above 15 m, the training data set performs better than the validation one (almost 3 m difference).



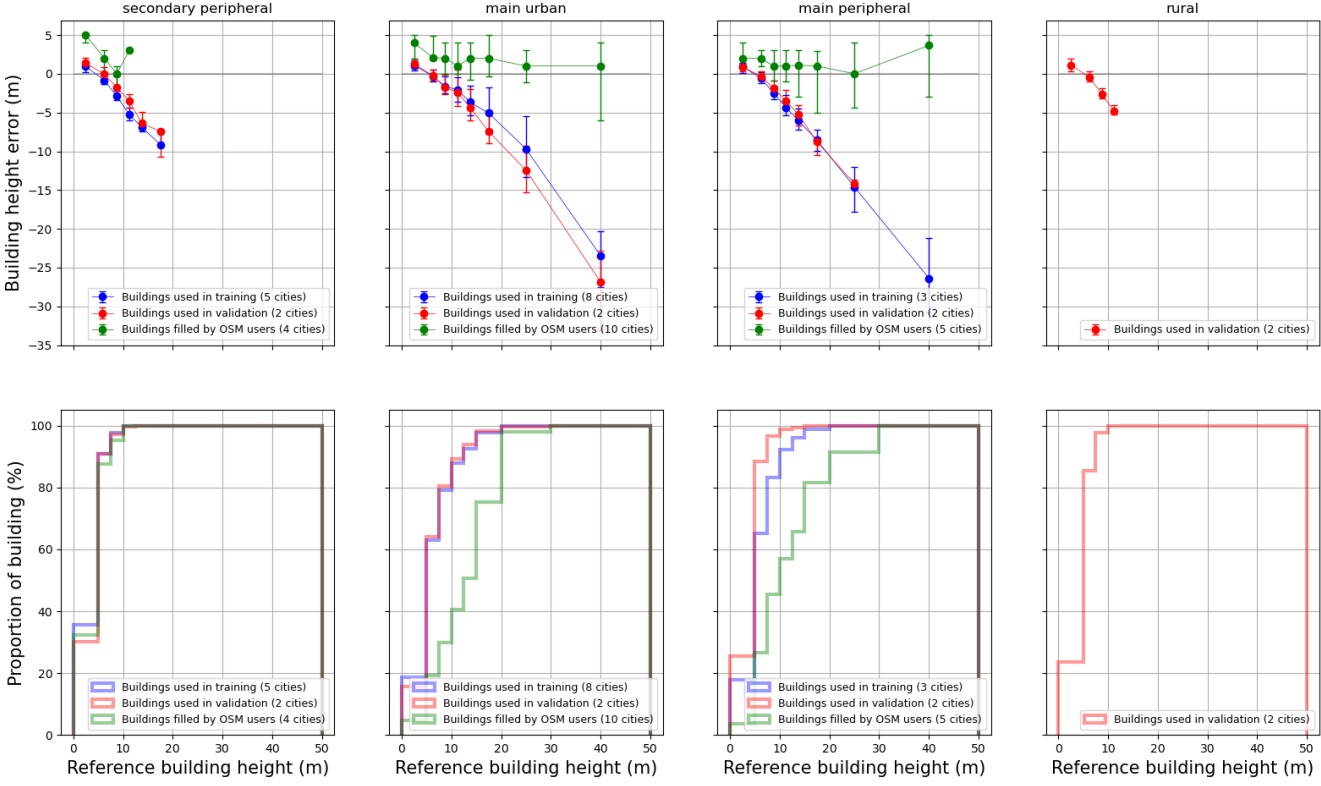

**Figure 7.** On the top: building height errors ($Err_{model}$ and $Err_{user}$) versus reference building height ($H_{OSM,true}$) for each type of city. The dot represent the median while the whiskers are the 1st and 3rd quartiles. On the bottom: cumulated distribution of reference building height ($H_{OSM,true}$) for each type of city. The interval used for the reference building height (the abscissa) are based on the following values: 0, 5, 7.5, 10, 12.5, 15, 20, 30, 50 m (values above 50 m are not considered since their number is negligible and they affect the reading).

This difference is attributed to Paris buildings data set: the two curves almost coincide if the latter is excluded. The reason is that Paris buildings are quite accurately calculated and represent a large part of the training data set (43.2% of the buildings higher than 15 m). The urban fabric (very dense block of buildings with courtyard) and the building heights are quite homogeneous in Paris, thus being well taken into account by the model. In most of other cities, a large amount of the high-rise buildings are isolated buildings (cf Figure 8 for an example with the city of Nantes). These buildings have probably very little shape or environment differences with smaller isolated buildings and are probably less numerous. Therefore, most of these buildings are seen as low-rise by the model.

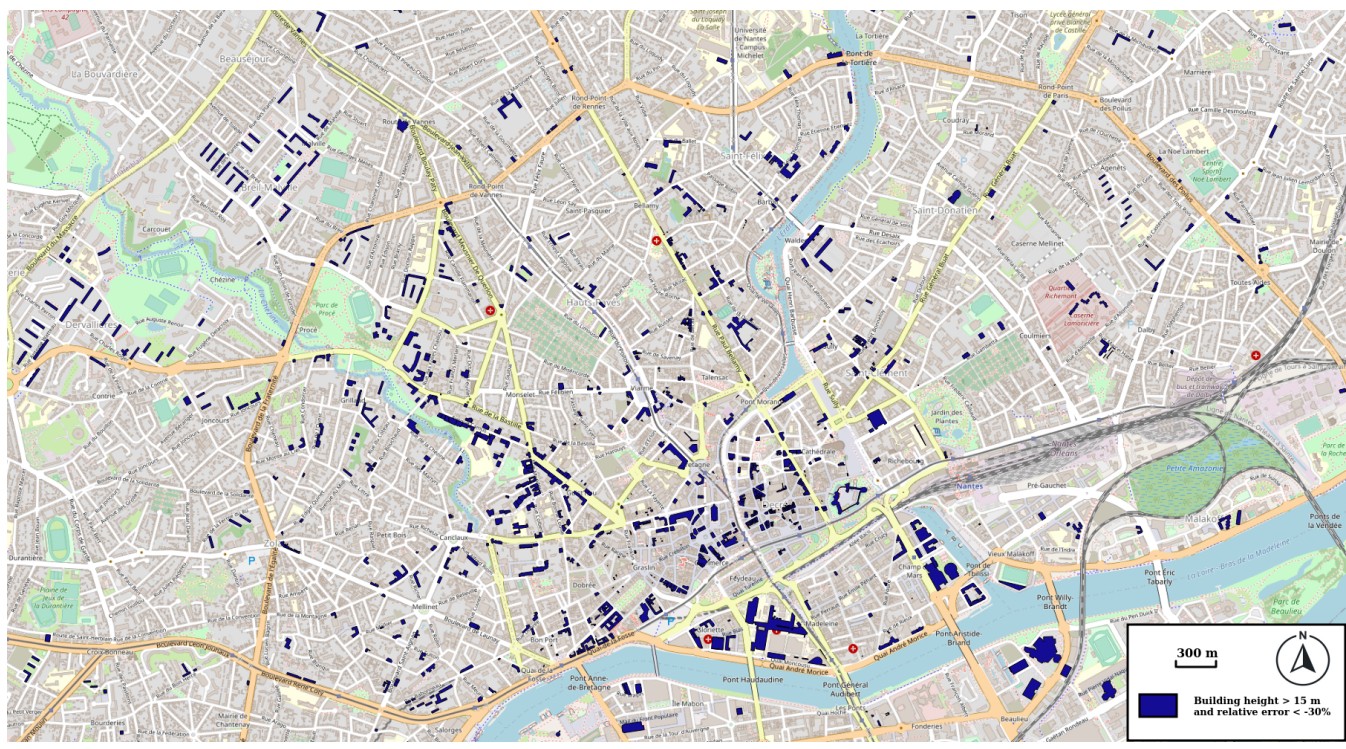

**Figure 8.** Buildings taller than 15 m for which the height underestimation is higher than 30%. ©OpenStreetMap contributors 2021. Distributed under the Open Data Commons Open Database License (ODbL) v1.0.

It is interesting to notice that the buildings already having a building height value in OSM (or at least a number of floors value) are most of the time slightly higher than the BDT ones (Figure 7). This is particularly the case for low-rise buildings and it may be explained by the fact that the BDT heights are taken at the lowest part of the roof. The OSM data can take into

account the roof height, which is most of the time equal to zero for tall buildings but non negligible for small buildings. The difference between height derived from OSM user filling and the reference data (BDT) is quite low for any building height. This result may be used to improve the model performance: when estimating the height of a given building, the random forest may take into account the height of a nearby building filled by an OSM user as an extra independant variable.

### 3.5  Spatial distribution of the building height at city scale

While most of the buildings higher than 15 m are underestimated, the model allows to represent well the spatial patterns of the third dimension: at grid scale, the average building height maps of estimated and reference values look quite similar (Figure 9 - Annex B for other cities). While the model slightly smooths the values, the city center, first ring and second ring are quite easily distinguishable. Note that this is not the case for cities having a more homogeneous spatial distribution of the building height values (e.g. Annecy which is constrained by the topography - cf Figure B1 in Annex B).





**Figure 9.** Results for the city of Nantes at grid cell (**a**) reference building height, (**b**) estimated building height, (**c**) absolute building height error and **d**) fraction of OSM buildings having height information. For case (**a**), (**b**) and (**c**), only cells having buildings and at least 90% of their buildings with no height value in OSM are displayed.

For most of the city pixels, the absolute error is under 2.5 m and only a small proportion of cells have an error higher than 5 m (Figure 9). This absolute error magnitude is within the accuracy of the reference building data set. Indeed, according to the data supplier (IGN) information (based on a 7,299,422 buildings sample), 8.2% of the buildings have an accuracy of 1 m, 13.5% an accuracy of 2.5 m, 68.8% an accuracy of 5 m while 9.5% have no accuracy information.



## 4   Conclusions

There is a need for a world-wide database of morphological indicators useful for many physical process interests (eg. parametric urban climate models, noise modelling, urban planning). The GeoClimate tool aims to tackle this issue using the OpenStreetMap data. However, most of the OSM buildings do not have any information concerning their height while it is a crucial parameter for urban climate studies. A Random Forest model has been integrated within Geoclimate to estimate the height of a building based on spatial indicators describing its shape, its relations to other buildings and the 2D characteristics of its close

environment.

This article presents the method to build and evaluate this model. The buildings from 14 French communes have been used to train the model while the evaluation was based on 8 French communes. Attention was paid to have as many types of territories as possible in the samples (based on the French definitions): main urban, main peripheral, secondary peripheral and rural.

The Random Forest model was tuned according to four parameters: the number of trees (best 350), the minimum size node

(best 0.01%), the maximum variables per tree (best 40%), the maximum leaf per tree (best 1100). The reference heights used for the training of our OSM buildings were based on a data set (French BDTopo - BDT) where buildings could not fit exactly with the OSM ones. Thus, the matching between each OSM building footprints and BDT buildings footprints has been quantified using the uniqueness value indicator. The latter equals to 1 if only one building from the BDT was used to feed the OSM building height, lower to 1 otherwise, best suited for the Random Forest model when values higher than 0.7.

Two communes (Paris and Meudon) demonstrate a specific behavior from the analysis. Appart of these, the median absolute error at cell scale was always lower than 1.6 m and 75% of the buildings or cells of any city had an error lower than 3.2 m. This level of magnitude is similar than the BDT data used for the training: 68.8% of the buildings heights demonstrate an uncertainty of 5 m.

Geographical indicators commonly used in urban climate studies have also been calculated at 100 m grid cell according to

the estimated building height. While the building height variability (standard deviation within a grid) is strongly affected by the building height estimation error (50% of the cells have more than 50% error in building height standard deviation value), the roughness length and sky view factor have a relative error of about 20% for 50% of the cells.

One of the major limitations of the model at the French scale is when applied to tall (>15 m) isolated buildings. However, it does not affect the recognition of the general patterns of a city: most of the high-rise buildings located in the center of cities

are quite well modeled although slightly underestimated.

In our opinion, while the data set resulting from the optimized Random Forest model is far from being perfect, it could be useful for climate analysis. We recommend a prior evaluation of what could be the effect of using the output of the RF model compared with the reference data usually employed by urban climate researchers. While we do not expect major differences when applied with parametric urban climate models at city scale, the spatial error might be quite high at neighborhood scale.

Thus, for researchers and practitioners willing to use Geoclimate at a finer scale (for example to automatically download landtype and land-use information for explicit modelling purpose), we recommend to contribute to the OSM project first. Specify the height of the most important buildings of their studying area in OSM can be done before running Geoclimate. At the end



of the day, they contribute to the improvement of the OSM data and they can freely take benefit from the Geoclimate tool. Concerning the building height modelling, the work may be continued by:

– evaluating the accuracy of the estimations using other reference data sets: in France it could be performed using more accurate reference data and in other countries with any existing reference data,

– improving the statistical modelling: (**i**) selecting a data set having a uniform distribution of building height as described Sect. 3.4, (**ii**) investigate other supervised methods, (**iii**) in the training data, get rid of OSM buildings having less than a certain fraction of BDT buildings covering them, (**v**) find more appropriate building properties which can be used as
independent variables (e.g. the height of the nearby buildings filled by OSM users) (**v**) identify a subset of the most appropriate variables in order to limit the adverse effects of noisy variables.

*Code and data availability.*  The major part of this work can be reproduced directly using the Software GeoClimate version 0.0.1 (the source code and executable file of this software version are permanently available on Zenodo at https://zenodo.org/record/6372337) and the scripts and data available on Zenodo at https://zenodo.org/record/6396361. GeoClimate downloads OpenStreetMap data using the overpass API
from the end point https://overpass-api.de/api, estimates building height when missing and calculates geographical indicators. The resulting data set presented in this paper have been obtained using the OpenStreetMap data between June and September 2021. It can be freely accessed at https://zenodo.org/record/6396361. The French BDTopo (version 2.2) is used only for training and evaluation purpose. It is a proprietary dataset provided by the French National Geographic Institute (IGN) and is available upon request. Thus it is unfortunately not possible to make this dataset freely accessible. This was one of the major motivation to perform this work, ie to create a methodology to automatically
create a topographic dataset containing buildings with estimated height.





## Appendix A: List of all spatial indicators used as independent variables

### A1    Building scale

| Indicator name | Indicator type | | | Name of the indicator in |
| --- | --- | --- | --- | --- |
| | Type and use | Form and size | Spatial relations | the GeoClimate documentation (version 0.0.1) |
| BUILD_TYPE | x | | | None (this is an input of GeoClimate) |
| BUILD_MAIN_USE | x | | | None (this is an input of GeoClimate) |
| BUILD_PERIMETER | | x | | PERIMETER |
| BUILD_AREA | | x | | AREA |
| BUILD_TOTAL_FACADE_LENGTH | | x | | TOTAL_FACADE_LENGTH |
| BUILD_COMMON_WALL_FRACTION | | | x | COMMON_WALL_FRACTION |
| BUILD_NUMBER_BUILDING_NEIGHBOR | | | x | NUMBER_BUILDING_NEIGHBOR |
| BUILD_AREA_CONCAVITY | | x | | AREA_CONCAVITY |
| BUILD_FORM_FACTOR | | x | | FORM_FACTOR |
| BUILD_PERIMETER_CONVEXITY | | x | | PERIMETER_CONVEXITY |
| BUILD_MINIMUM_BUILDING_SPACING | | | x | MINIMUM_BUILDING_SPACING |
| BUILD_ROAD_DISTANCE | | | x | ROAD_DISTANCE |
| BUILD_LIKELIHOOD_LARGE_BUILDING | | x | | LIKELIHOOD_LARGE_BUILDING |





## A2 Block scale

| Indicator name | Indicator type | | Name of the method in the GeoClimate documentation (version 0.0.1) |
| --- | --- | --- | --- |
| | Form and size | Aggregated statistics from lower scale | |
| BLOCK_BUILDING_DIRECTION_UNIQUENESS | x | | BUILDING_DIRECTION_UNIQUENESS |
| BLOCK_AREA | x | | AREA |
| BLOCK_BUILDING_DIRECTION_EQUALITY | x | | BUILDING_DIRECTION_EQUALITY |
| BLOCK_HOLE_AREA_DENSITY | x | | HOLE_AREA_DENSITY |
| BLOCK_CLOSINGNESS | x | | CLOSINGNESS |
| BUILD_AVG_PERIMETER | | x | None (average from lower scale) |
| BUILD_STD_PERIMETER | | x | None (standard deviation from lower scale) |
| BUILD_AVG_AREA | | x | None (average from lower scale) |
| BUILD_STD_AREA | | x | None (standard deviation from lower scale) |
| BUILD_STD_TOTAL_FACADE_LENGTH | | x | None (standard deviation from lower scale) |
| BUILD_STD_COMMON_WALL_FRACTION | | x | None (standard deviation from lower scale) |
| BUILD_STD_NUMBER_BUILDING_NEIGHBOR | | x | None (standard deviation from lower scale) |
| BUILD_AVG_AREA_CONCAVITY | | x | None (average from lower scale) |
| BUILD_STD_AREA_CONCAVITY | | x | None (standard deviation from lower scale) |
| BUILD_AVG_FORM_FACTOR | | x | None (average from lower scale) |
| BUILD_STD_FORM_FACTOR | | x | None (standard deviation from lower scale) |
| BUILD_AVG_PERIMETER_CONVEXITY | | x | None (average from lower scale) |
| BUILD_STD_PERIMETER_CONVEXITY | | x | None (standard deviation from lower scale) |
| BUILD_STD_MINIMUM_BUILDING_SPACING | | x | None (standard deviation from lower scale) |
| BUILD_AVG_ROAD_DISTANCE | | x | None (average from lower scale) |
| BUILD_STD_ROAD_DISTANCE | | x | None (standard deviation from lower scale) |
| BUILD_AVG_LIKELIHOOD_LARGE_BUILDING | | x | None (average from lower scale) |
| BUILD_STD_LIKELIHOOD_LARGE_BUILDING | | x | None (standard deviation from lower scale) |





## A3   TSU scale

| Indicator name | Indicator type | | | Name of the method in the GeoClimate documentation (version 0.0.1) |
|---|---|---|---|---|
| | Form and size | Planar density | Aggregated statistics from lower scale | |
| RSU_HIGH_VEGETATION_FRACTION | | x | | AREA_FRACTION_x |
| RSU_HIGH_VEGETATION_WATER_FRACTION | | x | | AREA_FRACTION_x |
| RSU_HIGH_VEGETATION_BUILDING_FRACTION | | x | | AREA_FRACTION_x |
| RSU_HIGH_VEGETATION_LOW_VEGETATION_FRACTION | | x | | AREA_FRACTION_x |
| RSU_HIGH_VEGETATION_ROAD_FRACTION | | x | | AREA_FRACTION_x |
| RSU_HIGH_VEGETATION_IMPERVIOUS_FRACTION | | x | | AREA_FRACTION_x |
| RSU_WATER_FRACTION | | x | | AREA_FRACTION_x |
| RSU_BUILDING_FRACTION | | x | | AREA_FRACTION_x |
| RSU_LOW_VEGETATION_FRACTION | | x | | AREA_FRACTION_x |
| RSU_ROAD_FRACTION | | x | | AREA_FRACTION_x |
| RSU_IMPERVIOUS_FRACTION | | x | | AREA_FRACTION_x |
| RSU_VEGETATION_FRACTION_URB | | x | | VEGETATION_FRACTION_URB |
| RSU_LOW_VEGETATION_FRACTION_URB | | x | | LOW_VEGETATION_FRACTION_URB |
| RSU_HIGH_VEGETATION_IMPERVIOUS_FRACTION_URB | | x | | HIGH_VEGETATION_IMPERVIOUS_FRACTION_URB |
| RSU_HIGH_VEGETATION_PERVIOUS_FRACTION_URB | | x | | HIGH_VEGETATION_PERVIOUS_FRACTION_URB |
| RSU_ROAD_FRACTION_URB | | x | | ROAD_FRACTION_URB |
| RSU_IMPERVIOUS_FRACTION_URB | | x | | IMPERVIOUS_FRACTION_URB |
| RSU_AREA | x | | | AREA |
| RSU_GROUND_LINEAR_ROAD_DENSITY | | x | | GROUND_LINEAR_ROAD_DENSITY |
| RSU_AVG_NUMBER_BUILDING_NEIGHBOR | | | x | AVG_NUMBER_BUILDING_NEIGHBOR |
| RSU_AVG_MINIMUM_BUILDING_SPACING | | | x | AVG_MINIMUM_BUILDING_SPACING |
| RSU_BUILDING_NUMBER_DENSITY | | x | | BUILDING_NUMBER_DENSITY |
| RSU_BUILDING_TOTAL_FRACTION | | x | | BUILDING_TOTAL_FRACTION |
| RSU_BUILDING_DIRECTION_EQUALITY | | | x | BUILDING_DIRECTION_EQUALITY |
| RSU_BUILDING_DIRECTION_UNIQUENESS | | | x | BUILDING_DIRECTION_UNIQUENESS |

## 305  Appendix B: Results for all cities

*Author contributions.*  Conceptualization: JB, EB, VM / Data curation: JB, EB, ELS / Formal analysis: JB, EB, ELS, FL / Funding acquisition: EB, ELS / Investigation: JB, EB / Methodology: JB, EB / Project administration: JB, EB / Resources: EB / Software: EB, JB, ELS, FL / Supervision: JB, EB / Validation: EB, JB, ELS, FL / Visualization: JB, EB / Writing - original draft preparation: JB, EB / Writing - review and editing: JB, EB, FL, ELS, VM

*Competing interests.*  The authors declare that they have no conflict of interest.





**Figure B1.** Results for the commune at grid cell (upper left panel) reference building height, (upper right panel) estimated building height, (lower left paner) absolute building height error and (lower right panel) fraction of OSM buildings having height information. For all panels except the lower right, only cells having buildings and at least 90% of their buildings with no height value in OSM are displayed.





**Figure B2.** Results for the commune at grid cell (upper left panel) reference building height, (upper right panel) estimated building height, (lower left paner) absolute building height error and (lower right panel) fraction of OSM buildings having height information. For all panels except the lower right, only cells having buildings and at least 90% of their buildings with no height value in OSM are displayed.



**Figure B3.** Results for the commune at grid cell (upper left panel) reference building height, (upper right panel) estimated building height, (lower left paner) absolute building height error and (lower right panel) fraction of OSM buildings having height information. For all panels except the lower right, only cells having buildings and at least 90% of their buildings with no height value in OSM are displayed.





**Figure B4.** Results for the commune at grid cell (upper left panel) reference building height, (upper right panel) estimated building height, (lower left paner) absolute building height error and (lower right panel) fraction of OSM buildings having height information. For all panels except the lower right, only cells having buildings and at least 90% of their buildings with no height value in OSM are displayed.





**Figure B5.** Results for the commune at grid cell (upper left panel) reference building height, (upper right panel) estimated building height, (lower left paner) absolute building height error and (lower right panel) fraction of OSM buildings having height information. For all panels except the lower right, only cells having buildings and at least 90% of their buildings with no height value in OSM are displayed.





**Figure B6.** Results for the commune at grid cell (upper left panel) reference building height, (upper right panel) estimated building height, (lower left paner) absolute building height error and (lower right panel) fraction of OSM buildings having height information. For all panels except the lower right, only cells having buildings and at least 90% of their buildings with no height value in OSM are displayed.







**Figure B7.** Results for the commune at grid cell (upper left panel) reference building height, (upper right panel) estimated building height, (lower left paner) absolute building height error and (lower right panel) fraction of OSM buildings having height information. For all panels except the lower right, only cells having buildings and at least 90% of their buildings with no height value in OSM are displayed.







**Figure B8.** Results for the commune at grid cell (upper left panel) reference building height, (upper right panel) estimated building height, (lower left paner) absolute building height error and (lower right panel) fraction of OSM buildings having height information. For all panels except the lower right, only cells having buildings and at least 90% of their buildings with no height value in OSM are displayed.







**Figure B9.** Results for the commune at grid cell (upper left panel) reference building height, (upper right panel) estimated building height, (lower left paner) absolute building height error and (lower right panel) fraction of OSM buildings having height information. For all panels except the lower right, only cells having buildings and at least 90% of their buildings with no height value in OSM are displayed.





**Figure B10.** Results for the commune at grid cell (upper left panel) reference building height, (upper right panel) estimated building height, (lower left paner) absolute building height error and (lower right panel) fraction of OSM buildings having height information. For all panels except the lower right, only cells having buildings and at least 90% of their buildings with no height value in OSM are displayed.





**Figure B11.** Results for the commune at grid cell (upper left panel) reference building height, (upper right panel) estimated building height, (lower left paner) absolute building height error and (lower right panel) fraction of OSM buildings having height information. For all panels except the lower right, only cells having buildings and at least 90% of their buildings with no height value in OSM are displayed.





**Figure B12.** Results for the commune at grid cell (upper left panel) reference building height, (upper right panel) estimated building height, (lower left paner) absolute building height error and (lower right panel) fraction of OSM buildings having height information. For all panels except the lower right, only cells having buildings and at least 90% of their buildings with no height value in OSM are displayed.





**Figure B13.** Results for the commune at grid cell (upper left panel) reference building height, (upper right panel) estimated building height, (lower left paner) absolute building height error and (lower right panel) fraction of OSM buildings having height information. For all panels except the lower right, only cells having buildings and at least 90% of their buildings with no height value in OSM are displayed.



**Figure B14.** Results for the commune at grid cell (upper left panel) reference building height, (upper right panel) estimated building height, (lower left paner) absolute building height error and (lower right panel) fraction of OSM buildings having height information. For all panels except the lower right, only cells having buildings and at least 90% of their buildings with no height value in OSM are displayed.





**Figure B15.** Results for the commune at grid cell (upper left panel) reference building height, (upper right panel) estimated building height, (lower left paner) absolute building height error and (lower right panel) fraction of OSM buildings having height information. For all panels except the lower right, only cells having buildings and at least 90% of their buildings with no height value in OSM are displayed.





**Figure B16.** Results for the commune at grid cell (upper left panel) reference building height, (upper right panel) estimated building height, (lower left paner) absolute building height error and (lower right panel) fraction of OSM buildings having height information. For all panels except the lower right, only cells having buildings and at least 90% of their buildings with no height value in OSM are displayed.





**Figure B17.** Results for the commune at grid cell (upper left panel) reference building height, (upper right panel) estimated building height, (lower left paner) absolute building height error and (lower right panel) fraction of OSM buildings having height information. For all panels except the lower right, only cells having buildings and at least 90% of their buildings with no height value in OSM are displayed.



**Figure B18.** Results for the commune at grid cell (upper left panel) reference building height, (upper right panel) estimated building height, (lower left paner) absolute building height error and (lower right panel) fraction of OSM buildings having height information. For all panels except the lower right, only cells having buildings and at least 90% of their buildings with no height value in OSM are displayed.





**Figure B19.** Results for the commune at grid cell (upper left panel) reference building height, (upper right panel) estimated building height, (lower left paner) absolute building height error and (lower right panel) fraction of OSM buildings having height information. For all panels except the lower right, only cells having buildings and at least 90% of their buildings with no height value in OSM are displayed.







**Figure B20.** Results for the commune at grid cell (upper left panel) reference building height, (upper right panel) estimated building height, (lower left paner) absolute building height error and (lower right panel) fraction of OSM buildings having height information. For all panels except the lower right, only cells having buildings and at least 90% of their buildings with no height value in OSM are displayed.



**Figure B21.** Results for the commune at grid cell (upper left panel) reference building height, (upper right panel) estimated building height, (lower left paner) absolute building height error and (lower right panel) fraction of OSM buildings having height information. For all panels except the lower right, only cells having buildings and at least 90% of their buildings with no height value in OSM are displayed.





**Figure B22.** Results for the commune at grid cell (upper left panel) reference building height, (upper right panel) estimated building height, (lower left paner) absolute building height error and (lower right panel) fraction of OSM buildings having height information. For all panels except the lower right, only cells having buildings and at least 90% of their buildings with no height value in OSM are displayed.





**Figure B23.** Results for the commune at grid cell (upper left panel) reference building height, (upper right panel) estimated building height, (lower left paner) absolute building height error and (lower right panel) fraction of OSM buildings having height information. For all panels except the lower right, only cells having buildings and at least 90% of their buildings with no height value in OSM are displayed.



*Acknowledgements.* The method presented in this paper has been integrated in the GeoClimate tool and developed within the following research projects:

– URCLIM (2017-2021), part of ERA4CS, a project initiated by JPI Climate and co-funded by the European Union under grant agreement No 690462

– CENSE (2017-2021), funded by the French National Research Agency (ANR) under grant agreement No Projet-ANR-16-CE22-0012

– SLIM (2020-2021), a Copernicus project C3S_432 Provisions to Environmental Fore- casting Applications (Lot 2)



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
