# Peer review of "Estimation of missing building height in OpenStreetMap data: a French case study using GeoClimate 0.0.1"

_Geoscientific Model Development, 2021_

## Referee Comment (RC3)

[referee-annotated manuscript omitted]

---

## Author Response (AR1)

**Detailed point by point response**
**to #*Anonymous reviewer 1**

Below are summarize each comment of reviewer 1: first the discussion between reviewer and authors concerning a specific point; second the changes made in the manuscript concerning this point (in this section, all lines refer to the marked-up manuscript version).

**Discussion 1**

Reviewer (R): The innovation compared to previous literature is minimal, in particular because the authors missed a study [NMD20] that looked at the very same question, with more data and a larger geographical scale. The authors should rethink the gap in the literature they are trying to address within their study by taking this closest reference point into account.

Authors (A): Thanks to *Anonymous Referee #1* for this remark and the very interesting study he pointed out which is indeed quite similar to our work. Even thought we did not have knowledge of this study, we think that at least three points make our work useful anyway:
- The geographical indicators used as explicative variables are different between [NMD20] and our study, however results tend to be similar, which in a sense is worth of being investigated.
- While the geographical scale is larger and the investigation much deeper than ours, [NMD20] work seems hard to replicate (and thus hard to use) since their code seems only partially available (https://gitlab.pik-potsdam.de/nikolami/learning-from-urban-form-to-predict-building-heights). We think that the strength of our work (and this is why we chose GMD to publish our work) is its accessibility and its possibility of reuse.
- The building height estimation accuracy has direct consequences on spatial indicators used to model / analyze urban climate. Nothing is

related to this topic in [NMD20] while these consequences are investigated in our work.

R: I agree that the aspect of replication of previous research in another geographical context + methods easier to use for practitioners are valuable. I am not sure i understand the third point.

A: The third point was related to the spatial indicators calculated at 100m grid cell (average building height, sky view factor, etc. ) using the building height. This information is important for the urban climate community since most of the regional scale atmospheric models consider average building values at grid scale. The accuracy of the averaged building height is slightly improved when compare to the accuracy at building scale (even thought the accuracy gain is lower than what we expected).

➢ Changes in the manuscript : We have updated our manuscript from l. 52 to 68 to take into account the [NMD20] study and also better highlight the contribution of our work to the field.

**Discussion 2**

R: There is a lack of framing regarding under which context the method should be used: e.g. is it to predict in areas with no data available as proposed in [NMD20], which requires to test the spatial generalization of the model, or to fill the gaps in a city where data is available as in [BIL17] and where more over-fitting makes sense? Those different cases require different training and testing approaches. It seems that the authors are primarily interested in the first one, if so, the methods should more robustly assess generalization, see next point.

A: The main objective of our work is actually two-fold:

- Proposing a whole methodology to estimate building height from Free and Open Source data using a Free and Open-Source Software (FOSS) available online at https://github.com/orbisgis/geoclimate/wiki ,

- Evaluating the performance of this methodology in an area where we can easily obtain data as French researchers.

However, as *Anonymous Referee #1* guessed it, the mid-term objective of our work is to verify that the methodology applied in France is still valid in other countries. This is why we have designed GeoClimate to be simple to use for anyone in any country. Any researcher with an access to local data is able to assess the performance of the current model for his country or to use GeoClimate to create a new model (might be the one developed by [NMD20]) which would be more appropriate for his country.

R: I still believe the authors could do a better job explaining how different use cases of missing data (e.g. partial unavailability within a city vs no data available in a region) require different training approaches, e.g. see:

- Meyer, Hanna, et al. "Importance of spatial predictor variable selection in machine learning applications–Moving from data reproduction to spatial prediction." Ecological Modelling 411 (2019): 108815.
- Le Rest, Kévin, et al. "Spatial leave-one-out cross-validation for variable selection in the presence of spatial autocorrelation." Global ecology and biogeography 23.7 (2014): 811-820.

A: We are not sure to understand this point. We have designed a method to estimate building height using OSM data. If the users want to apply this method to a specific area where the amount of data available is limited or absent (few or no buildings, few or no road, few or no vegetation patches, etc.), we first recommend the users to contribute to the OSM project (for example using the open and collaborative project "Missing Map"). If they want to train the algorithm using his own reference data, they should also contribute to OSM first since otherwise the model created will probably have low quality. We have added a short comment about this point in the conclusion of the manuscript.

➢ Changes in the manuscript : We have added a short comment about this point in the conclusion of the manuscript (l. 307 to 309).

**Discussion 3**

R: The methodology needs improvements to be more robust. The results seem reasonable e.g. the RMSE and MAE values, but the training and test procedure should be improved. Some examples of points I believe could be addressed: First, the authors should undertake cross-validation to test their model on different folds of the data to account for different urban situations that easier or hard to train/predict on. Second, spatial cross-validation on spatially-distant folds would be particularly relevant to enhance/demonstrate generalization. Third, why choosing train and test cities in the same region e.g. Corbonod and Annecy or Nantes and Saint-Nicolas-.. are close, while there are so many cities in France to separate further spatially the sets?

A: It seems that this point is viewed by *Anonymous Referee #1* as a major methodological concern. In the following, we try our best to answer each of these points.

We are not sure to understand well the differences between your second and your third points. We try to clarify what we have done in the training and validation stages:

- Concerning the training, we have actually trained the model on 70% of the training data and validate it on the 30% remaining (cf. section 2.2.3 and Figure 5). This has been done using all cities of the training dataset. The objective was to identify what was the best set of parameters for the RandomForest model.
- Then we have applied this "optimized" model on the validation dataset which consists of spatially distant cities from the ones used in the training. Thus if the trained model would have been poorly designed, it would have led to worse results using the validation cities. Your third point mentions the distance between training and validation cities which is not sufficient large. We do not think that using cities for validation "more remote from the training ones" will modify the performance of the model. Many reasons leads to this assumption:
  - We have actually tested a previous version of the model on more remote cities and the performance was similar,

- We think that the city's topography (mainly water bodies and mountains), the city's historical heritage (e.g. periods of demographic expansion) and how cities are located within the attraction cluster such as defined by the French INSEE (types defined Table 1) plays an important role on the city's morphology. From this perspective, Corbonod (validation dataset) is located within a mountainous area (such as Annecy or La Thuile - training) but it is not at all affected the same way by the cluster attraction (rural area versus urban areas respectively) since its expansion is much less constrained by the mountains. The other cities in the region (Dijon, Charnay-lès-Mâcon and Pont-de-Veyle - validation) are located in much flater areas. Concerning West France, Nantes (training) is a main urban area built along a large river (the Loire) while the closest cities in the validation dataset are cities within or outside a much smaller attraction cluster (Redon).
- [NMD20] has obtained almost no improvement of their prediction when they add to the training dataset a city which is really close to the ones used for validation. In the meantime, [NMD20] has obtained a much better improvement adding randomly chosen local data: "*In Experiment 2, we added 2% of local data to the training set data which resulted in noticeable accuracy gains compared to Experiment 1 for both test sets. In contrast, Experiment 3 where we added Berlin to the training set for predicting Brandenburg did not noticeably improve the results.*" (p. 12). [NMD20] also mentions that "*the townhouses that are so typical for Berlin are not as common even in large cities in Brandenburg, despite the geographical vicinity.*" (p.14).

R: My point was not that you may have poorly designed the training set resulting in too pessimistic prediction, but rather than there are ways to more robustly ensure that the choice of training and test set does not generate over-optimistic results (as in, depending on what you are using the results for, e.g. to say that one can expect that the accuracy of the inferred data across the French territory, where there is no ground truth, will be X, Y, Z). *"We have actually tested a previous version of the model on more remote cities and the performance was similar"* -> why not including this?. Thanks for the interesting discussion on topography etc. The results you mention from [NMD20] might be case-specific and not generalizable, so i would be careful with basing your intuition on these here.

A: We cannot include the cities we have tested before because some of the explaining variables have been modified since this date (it was a previous version of the model). But we have added the sentence in section 3.2 of the article as recommended by *#Anonymous referee 1*. Concerning the argument about [NMD20] result, right we would not have drawn any conclusion solely from this result (since we did not know about this study when we first designed our work). However, we thought it was an additional argument to think that proximity is not necessarily equal to spatial similarity.

> ➢ Changes in the manuscript : As said in our last answer, we have added the sentence recommended by the reviewer in section 3.2 (l. 218-219)

**Discussion 4**

R: The text could be much clearer. In particular, I found the structure of the introduction confusing (this relates to the previous points on lack clear gap in literature, use case, etc.). The presentation of the results is also perfectible. Some metrics are given without clear indications of the sets they are referring to e.g. are the RMSE line 170 and "all cities" line 175 for the training or test set, or both? Results from training and test should in principle be presented separately, not together as on Fig. 6, as they represent different prediction settings. One/few summary result table(s) would also help the reader.

A: The introduction has been modified for clarification purpose (cf. the "diff" version of the attached pdf file).

In the preprint, there is no RMSE value line 170, and the words "all cities" are not mentioned line 175. Could the reviewer cite the paragraph or the section that he suggests to improve ?

Concerning Fig. 6, training and validation were on purpose shown on the same axis to show how similar is the error for both of them. However, for more clarity we have also summarized Fig. 6 and 7 in tables as recommended by *Anonymous Referee #1*.

R: Thanks for adding the tables.

➢ Changes in the manuscript : We have updated the introduction in order to make clearer the literature gap and our contribution to the field (l. 52 to 68) and added two Tables to better summarize the results (Table 5 p. 15 and Table 6 p. 18).

**Discussion 5**

R: Why using OSM if you have higher quality data from a government source? BDTOPO is great in France, why using data with uncertain coverage when best coverage is available? There is also a lot of great government data across Europe, so why OSM specifically? For scaling globally? Because GeoClimate is specifically built for OSM? I believe this is not explained. Also, one would need to take into account that in areas where OSM building footprint coverage is low, say rural Greece, the model will likely be wrong as the urban form input will be wrong. If the goal of the authors is to specifically investigate prediction from OSM, then an option to differentiate this study from [NMD20] could be to predict for different scenarios of OSM quality and compare the results, which might show that OSM is good enough even with medium-low quality, or not, and then identify where and why?

A: Concerning the debate about the use of OSM database instead of government data, there are several arguments in favor of the first:

1. When we started the project, OSM was the unique open data set available on the whole French territory (the BDTopo V3 is free and open access but the V2.2 is not).
2. OSM covers the whole planet.
3. OSM gives free and unlimited access to the entire database, with a complete history of changes.
4. OSM provides easy data access thanks to the Overpass API that permits to download data on demand for any part of the territory (using a bbox, a name for a commune...).
5. OSM data model is flexible (thanks to the tags approach) and can quickly be updated by any people in the world

6. And since it's open, anyone can also help improve the quality : edit the geometry or add new descriptors.

One of our main objectives is to provide a methodology and an open tool to produce climate and environmental indicators for any communities (e.g. geographers, urban stakeholders, environmental and climate specialists), therefore we believe that the OSM source was the best option. It is now possible for anyone having local government database to compare them to the building height estimated within GeoClimate.

R: Thanks for the explanations. I believe it would still be important to explain the issues when using OSM for predictive features, in particular missing building footprints / land use polygons / etc. in many areas that result in biased urban form and consequently wrong input feature values. Ideally this would be something the authors could investigate as previously suggested, by looking by artifically creating missing data situations, but I am happy with at least a mention of this issue. I believe this is important for the users of your model to have in mind the limitations of OSM so that they make the appropriate analysis beforehand. By the way, i do not see any completeness analysis of OSM in your manuscript. Quickly checking simple metrics like accordance between total footprint area in OSM and BDTOPO for the selected cities would be good.

A: Thank you for this really interesting comment which clearer some of the previous #Anonymous referee 1 comments. The results we show in this manuscript actually considers that the OSM data coverage is homogeneous within the French territory while missing informations (buildings, vegetation, roads, etc.) may indeed cause estimation bias. We have calculated simple metrics to illustrate the good correspondance between OSM and BDTopo: for all studied cities, the building fraction is higher in OSM than in BDTopo (even if this difference do not indicate a degree of completeness since both data set contain their own limitations). We have added a short paragraph to adress the missing data sensitivity analysis as potential future work.

➢ Changes in the manuscript : We have updated the conclusion (l. 307 to 309, l. 319 to 320 and l. 324 to 325) to better discuss the

limitations of the current work due to OSM data and to adress some more future works.

**Detailed point by point response**
**to *#Anonymous reviewer 2**

Below are summarize each comment of reviewer 2: first the discussion between reviewer and authors concerning a specific point; second the changes made in the manuscript concerning this point (in this section, all lines refer to the marked-up manuscript version).

**Discussion 1**

Reviewer (R): Introduction - mention all global open source and/or closed datasets you know about that have data on building heights (in addition to OSM).

Authors (A): We do not know any open source or closed data set having building height information at world scale. This problem is actually well described in Masson et al (2020) as we say line 31 in the article *"However, information concerning the vertical dimension is rarely available (Masson et al., 2020)"*. But *Anonymous Referee #2* is right, the location of the sentence (between two paragraphs describing OSM data) is not perfect or the sentence not accurate enough (we might think this lack of vertical dimension is only an OSM issue while in Masson et al. (2020) it is described as a global issue for any data set). We have slightly modified the sentence in order to make it more understandable that it is a missing information in all datasets.

> ➢ Changes in the manuscript : The sentence concerning the lack of building height information at building scale has been modified in order to better describe that it is a global statement true for all data and not only limited to OSM data (l. 31 to 32).

**Discussion 2**

R: Data and methods - comments are added in the attached file in order to further clarify the methodology. For example, what was the rationale behind selecting the specific study areas for training and validation. Why did you use only four types of spatial indicators out of 62 indicators? Please elaborate a bit.

A: The study areas selected have been chosen according to the following criteria. The data sets (both training and validation) need to contain cities:
- relatively far from each other to have different history / cultural construction heritages,
- Having different geographical contexts (near mountain, near the sea or far from both)
- of different types (according to the INSEE definition – cf. Table 1)

We have described with more detail these informations in the manuscript. Answer concerning indicators comes below when answering to pdf annotations.

➢ Changes in the manuscript : We have described with more details the method used to choose the study areas (l. 102 to 108).

**Discussion 3**

R: Results - for which urban class did you obtain the best / worst estimate of building height? Elaborate this more clearly.

A: As already stated in the manuscript, there is no clear performance increase / decrease for a specific urban class. However, #Anonymous Referee 2 is not the only one to ask for clearer results since reviewer #Anonymous Referee 1 was also asking for a Table to better summarize the results. Thus we have added two tables (Please refer to Table 5 and Table 6) in this perspective and slightly modified the results description section (cf. supplement material enclosed to our 1st answer to #Anonymous Referee 1).

> ➢ Changes in the manuscript : We have added two Tables to better summarize the results (Table 5 p. 15 and Table 6 p. 18).

**Discussion 4**

R: Eng language comments are also added.
A: Thank you for the English language modifications.

> ➢ Changes in the manuscript : Few language typo have been modified according to reviewer 2 comments (l. 112, 154).

**Comments found in the pdf annotated by referee 2**
**Comment 1**

R: *"each building and its environment" (p.3):* How is the environment of the building defined? What is the size of it? Does it differ depending on building size?
A: Good catch. The building environment is actually defined by the limit of the Topological Spatial Unit it belongs to (cf. Figure 3). This information was missing at this stage, we have added it in the manuscript.

> ➢ Changes in the manuscript : Definition of the building environment added in the manuscript (l. 87 to 88)

**Comment 2**

R: *"Table 4" (p. 8):* Why only these four out of 62 indicators? Please elaborate a bit.
A: Table 4 presents the main categories of spatial indicators, not each indicators. The table containing the list of the 62 indicators is in Annex A. We thought that Table 4 was a good summary of the types of indicators used as explaining variables and was better than the full list.

➢ Changes in the manuscript : No change in the manuscript

**Comment 3**

R: *"reference height (actually if the user fills only the number of storey a simple rule is used to calculate the building height)" (p. 10):* Specify the rule here
A: Good point. The rule is
Building height = number of storey * storey height

By default, storey height is set to 3 m. Even thought this value may vary quite a lot between construction age and building type (see Biljecki et al. (2017) Figure 5),  it seems a reasonable value according to the one observed in the literature (ranging from 2.8 and 3.5 m – Biljecki et al. (2017) section 2.2.1).

➢ Changes in the manuscript : We have added the above informations in the manuscript (l. 178 to 189).